# Comparing Non-Thermal Plasma and Cold Stratification: Which Pre-Sowing Treatment Benefits Wild Plant Emergence?

**DOI:** 10.3390/plants12183220

**Published:** 2023-09-10

**Authors:** Vilma Gudyniene, Sigitas Juzenas, Vaclovas Stukonis, Vida Mildaziene, Anatolii Ivankov, Egle Norkeviciene

**Affiliations:** 1Lithuanian Research Centre for Agriculture and Forestry, Instituto al. 1, Akademija, LT-58344 Kedainiai, Lithuania; vaclovas.stukonis@lammc.lt (V.S.); egle.norkeviciene@lammc.lt (E.N.); 2Vilnius University Life Sciences Centre (VU LSC), Institute of Biosciences, Sauletekio Av. 7, LT-10257 Vilnius, Lithuania; sigitas.juzenas@gf.vu.lt; 3Faculty of Natural Sciences, Vytautas Magnus University, Vileikos Str. 8, LT-44404 Kaunas, Lithuania; vida.mildaziene@vdu.lt (V.M.); anatolii.ivankov@vdu.lt (A.I.)

**Keywords:** native plants, emergence, non-thermal plasma, cold stratification, pre-sowing treatment

## Abstract

Meadow restoration and creation projects have faced a lack of local seed diversity due to the limited availability of seed sources. Non-thermal plasma technologies are being developed for agriculture and do not cause damage to heat-sensitive biological systems. This technology has shown the potential to improve agronomic seed quality by enhancing germination and promoting plant growth. However, there is almost no information about the effect of non-thermal plasma pretreatment on the seedlings’ emergence of wild plant species. Therefore, this study aimed to evaluate the effect of non-thermal plasma on the emergence of 17 plant seeds originating from local meadows in Lithuania and compare it with the cold stratification pretreatment. The results obtained indicate that there were differences in emergence parameters among the species. However, NTP did not show statistically significant differences from the control. Non-thermal plasma improved the kinetic parameters of emergence for a few specific species’ seeds, such as *Anthyllis vulneraria* and *Prunella grandiflora*, while the cold stratification pretreatment enhanced emergence for a broader range of plants. Significant differences were observed between non-thermal plasma and stratification pretreatment, as well as between the control and stratification groups. Both methods also had a negative impact.

## 1. Introduction

The sowing of native wild plant seed mixtures, composed of target meadow habitat species, is used to create urban greenery [1,2] and reinvent urban ecosystems [3,4], making a positive impact on city biodiversity [5,6], or restore natural meadow habitats [7]. Fast and reliable germination, low diversity, and quick establishment of wildflower seeds are essential indicators that help select suitable plants for such meadow mixture designs [8]. This approach aims to maintain the natural genetic structure of the species’ populations [9]. However, these mixtures are rarely composed of locally grown plant species seeds, and even natural meadow restoration is impacted because of the limited availability of seed sources [10,11]. Furthermore, in many countries, wildflower seeds are not yet cultivated on a large scale, and the supply does not meet the growing demand for such wild plant seeds. The development and testing of new methods to improve seed germination and emergence are therefore key to promoting wider wild plant cultivation.

In this study, we aim to address the question of how innovative technologies, such as physical stressors like non-thermal atmospheric pressure plasma (NTP), can improve the seedling emergence of perennial meadow plants. We additionally conducted a comparative analysis between the outcomes of seed pretreatment using NTP and the effects of cold stratification on seedling emergence and leaf length.

Cold stratification is widely employed to enhance the germination of wild plant seeds [12,13,14,15,16]. The non-thermal plasma method has been extensively explored in recent decades to improve the germination parameters of cultivated plant seeds [17,18,19,20]. Many studies have shown that NTP can improve the germination parameters. The germination and vigor indexes of various crops, as well as the early growth of seedlings and seed production in cultural plants, are significantly positively affected by NTP [21,22,23,24,25,26,27]. For instance, a short-time plasma pretreatment (1 min) promoted *Arabidopsis thaliana* seed germination and seedling growth [28]. Germination tests in vitro of red clover cultivar ‘Vyčiai’ (*Trifolium pratense*) revealed that NTP pretreatments increased germination rates by up to 24% [29]. It has conclusively been shown that NTP pretreatment can stimulate legume root nodulation and result in positive changes in seedling agronomic performance (like biomass production) [30]. NTP-treated tomato seeds exhibited faster germination rates and improved parameters of seedling growth [31]. NTP pretreatment significantly enhanced germination characteristics, vigor index, and seedling length of *Daucus carota* seeds [21]. Short plasma exposure to NTP increased the mean germination rate for wheat seeds [32]. Moreover, recent research has revealed that the effects of NTP are more persistent and complex than previously thought [30].

Most of these reports represent studies on seed germination or seedling emergence and early growth (from 4 days to several weeks). However, the emergence of wild plant seedlings is extended over time and longer-term studies are needed to properly assess the observed effects. 

Furthermore, there is only a single publication on the pre-sowing effects of NTP on wild seedling growth and plant seed production. In the study undertaken by Liu et al. [33], both direct and indirect NTP effects on the germination of the *Dianthus* genus were investigated; however, the results did not indicate a germination boost after an indirect pretreatment (watering with plasma activated water) and there was no significant difference between the treated and control groups in the direct pretreatment with NTP. The germination of an endangered native Iranian species *Salvia leriifolia* was improved after the short exposure of seeds to NTP [34]. Such finding indicates the potential of NTP pretreatments to be used for the stimulation of the germination of wild plant seeds, and further studies in this area are required.

In this study, 17 species of wild plants were selected for investigation. These species are primarily common in grassland habitats [35], with the majority belonging to three distinct phytosociological classes found in the region: *Festuca-Brometea* (*Anthyllis vulneraria*, *Dianthus borbasii*, *Filipendula vulgaris*, *Helianthemum nummularium*, *Salvia pratensis*, *Scabiosa columbaria*, and *Scabiosa ochroleuca*), *Molinio-Arrhenatheretea* (*Polemonium caeruleum*, *Ranunculus acris*, and *Rhinanthus serotinus*), and *Trifolio-Geranietea* (*Campanula glomerata*, *Veronica teucrium*, *Vicia pisiformis*, and *Prunella grandiflora*). *Campanula glomerata* and *Galium boreale* are characterized as species with wide ecological amplitudes, commonly found in a broad range of grassland habitats [35,36]. Among them, there are six rare and protected plant species in Lithuania: *C. bononiensis*, *D. borbasii*, *P. caeruleum*, *P. grandiflora*, *S. pratensis*, and *V. pisiformis* [37]. 

Among all species, three germination strategies emerged: some species postponed germination until after a period of cold, winter-like temperatures, indicating physiological and/or morphological dormancy mechanisms [38,39]. Table 1 shows an overview of the results from conducted experiments from the Seed Information Database ENSCOBASE [39]. Certainly, within a species, consistent results were not always obtained in different experiments. However, we have opted to report the highest recorded percentage of germination achieved for the tested species (or closely related ones), along with the corresponding germination duration in days. In cases where specific germination times were not provided, we have chosen to focus on experiment reports that include the mean time to germinate (MTG) index. Data analysis shows that our studied plants, *A. vulneraria*, *D. grandiflora*, *P. grandiflora*, *S. pratensis*, and *P. caeruleum*, are quite rapidly germinating species. Nevertheless, pre-sowing treatments are advised for *C. glomerata*, *G. boreale*, *P. caeruleum*, *V. pisiformis*, *H. nummularium*, and *R. serotinus* to overcome dormancy and attain a shorter germination period. There is a lack of data regarding the emergence and pre-sowing treatment recommendations for *C. bononiensis*, *P. grandiflora*, *R. acris*, *R. serotinus*, *S. ochroleuca*, *V. teucrium*, and *V. pisiformis* (Table 1).

The present research offers important insights into understanding the emergence of seedlings of the tested species, thus providing new insights in developing more effective approaches to promote their successful establishment and growth in meadow restoration and creation projects.

## 2. Results

### 2.1. Seedling Emergence Analysis

To reveal the temporal progression of seedling emergence, data collected from four separate replications for each of the three treatments within each of the 17 species were utilized to calculate cumulative probability functions (CDFs). This computation was carried out using nonparametric maximum likelihood estimators (NPMLEs) owing to their capacity to offer a more comprehensive analytical approach for interval-censored data, as compared to relying solely on a set of germination indices. Moreover, these nonparametric time-to-event models have the potential to uncover biologically significant parameters, specifically efficiency, rate, and uniformity, thus encompassing essential facets of seedling emergence. Figure 1 displays the P*_perm_*) levels following a Wilcoxon exact test conducted to assess the equality of time-to-event curves. Significant differences (P*_perm_* < 0.05) in seedling emergence trends were observed across different seed pretreatments for eight species: *A. vulneraria*, *D. grandiflora*, *F. vulgaris*, *G. boreale*, *H. nummularium*, *P. caeruleum*, *R. acris*, and *R. serotinus* (Figure 1). Upon analyzing the graphs in Figure 1, the most noticeable upward shifts in NTP curves compared to the control group were detected for *A. vulneraria*. There was a consistent and uniform increase recorded until the end of the third week of the experiment (Figure 1). A marked difference in the trend of seedling emergence was observed between NTP pretreatment and stratification for *F. vulgaris*. Upon comparing the control group to the stratification pretreatment, it became evident that the seedling emergence curve was notably higher for both *R. acris* and *H. nummularium*. However, specifically within the stratification curve, the dynamics of emergence shifted from rapid to virtually no new seedlings appearing by the 10th day of the experiment. Similar seedling emergence patterns after the stratification pretreatment were recorded for *G. boreale* and *P. caeruleum* in comparison with the control group (Figure 1).

When the rank tests for NPMLE methods were uncertain due to intersection (*P. grandiflora*, *D. borbasii*, *C. glomerata*, *V. teucrium*, and *S. columbaria*; P*_perm_* ≥ 0.05), the analysis included the Cramér–von Mises-type distance statistic for Kernel Density Estimates (KDEs) [40]. By employing this nonparametric model and test, we exclusively identified notable distinctions in seedling emergence patterns for *P. grandiflora*. Additionally, upon evaluating the nonparametric time-to-event curves (NPMLEs) for *P. grandiflora* (Figure 1), a discernible increase in the seedling emergence percentage was observed after the NTP pretreatment, in contrast to the control group. Under the NTP pretreatment, the highest speed of seedling emergence occurred between the 4th and 11th days, and then remained at a steady level after the 15th day until the end of the experiment. While both seed pretreatments resulted in an accelerated initiation of seedling emergence, the impact of the stratification on emergence was considerably diminished by the onset of the second week of the experiment, as compared to the control group (Figure 1). However, the gradual trend of emergence continued for a longer period, indicating that after the stratification pretreatment, *P. grandiflora* sprouts should be observed for an extended period. 

This study revealed that seed pretreatments had a significant effect (P*_perm_* < 0.05) on seedling emergence for 52.9% of the species that were tested. Further investigation, with a more comprehensive focus on the analysis of emergence percentages after the 14th (EP_14_) and 28th (EP_28_) days, as well as the T_10_ and T_30_ indices of the speed of species’ seedling emergence, is presented in Table 2.

Considering the statistically significant differences in the average seedling emergence after 14 days due to seed pretreatment compared to the control, a more pronounced effect was observed with the stratification rather than with NTP. An important finding was that NTP caused differences in EP_14_ for 23.5% of the tested species, whereas stratification had an impact on 29.4%. 

NTP showed an increase in EP_14_ (*p* < 0.05) for *P. grandiflora*; in contrast, there was a significant (*p* < 0.05) decrease in the average EP_14_ observed for *S. columbaria*, *D. grandiflora*, and *R. acris.* The stratification showed a significant increase in EP_14_ (*p* < 0.05) for *C. bononiensis*, *G. boreale*, *H. nummularium*, and *P. caeruleum*; in contrast, there was a significant (*p* < 0.05) decrease in the average EP_14_ observed for *F. vulgaris* (Table 2). The statistically significant differences in average seedling emergence after 28 days diminished when comparing the effect of seed pretreatment to the control. The tests identified that NTP caused variations in EP_28_ for 5.9% of the tested species, whereas stratification had an impact on 23.5%. NTP showed a higher increase in EP_28_ (*p* < 0.05) only for *A. vulneraria*. Stratification showed a higher increase in EP_28_ (*p* < 0.05) for *G. boreale*, *H. nummularium*, and *P. caeruleum*, whereas a significant (*p* < 0.05) decrease in the average EP_28_ was observed for *F. vulgaris* (Table 2).

When comparing both pretreatments (NTP vs. stratification), it was found that NTP resulted in a statistically significant increase in EP_14_ and EP_28_ for only 11.8% of the tested species, whereas stratification had a positive impact on 41.7%. Additionally, as a result of seed pretreatment, there were 3.5-fold more species that exhibited an increase or decrease in EP after 14 days compared to after 28 days.

A sole statistically significant positive effect (*p* < 0.05) was discerned for the EP_14_ of *P. grandiflora* under the NTP pretreatment, demonstrating a 1.4-fold increase when compared to the control group (Table 2). Nevertheless, no statistically significant difference between EP_28_ values in different pretreatments was found. Furthermore, it is noteworthy that the time needed to attain 10% seedling emergence (T_10_) under NTP pretreatment was significantly decreased by 1.5-fold in comparison to the control for *P. grandiflora*.

In the present study, comparing the seedling emergence pattern of *A. vulneraria*, statistically significant differences were observed among seed pretreatments (Figure 1, Table 2). Despite a higher average EP_14_ observed in the NTP pretreatment after 14 days, none of the differences in comparison with the control or stratification were statistically significant. However, a substantial and significant positive increase (*p* < 0.05) in EP_28_ was evident (Table 2), indicating a 4.5-fold increase compared with the control.

A statistically significant decrease (*p* < 0.05) in EP_14_ for *R. acris* was observed under the NTP pretreatment, resulting in a 1.3-fold reduction compared to the control (Table 2). However, there was no statistically significant distinction in EP_28_ values between NTP and the control. Nevertheless, the difference in EP_28_ between pretreatments (NTP vs. stratification) indicated a 1.3-fold increase for stratification. Also, stratification led to a faster emergence of seedlings, bypassing both the control and NTP treatments (T_10_ and T_30_ values) (Table 2). The values of T_10_ for *R. acris* under stratification were significantly (*p* < 0.05) lower by 1.6-fold compared to the control and 1.7-fold compared to NTP. Similarly, T_28_ under stratification was significantly (*p* < 0.05) decreased by 1.9-fold compared to the control and 2.0-fold compared to NTP (Table 2). A similar and significant difference in EP_14_ was recorded for *S. columbaria*: a 2.1-fold decrease compared to the control. The values of T_10_ indicated that stratification resulted in a quicker emergence of *S. columbaria* seedlings until the 14th day, similar to *R. acris*, surpassing both the control and NTP pretreatments 1.2-fold and 1.8-fold, respectively. One unanticipated finding was that the NTP pretreatment had the most substantial adverse impact (*p* < 0.05) on the germination of *D. grandiflora* seedlings, resulting in a significant 10.0-fold decrease in EP_14_ compared to the control (Table 2).

Stratification showed a statistically significant increase in EP_14_ (*p* < 0.05) for *C. bononiensis* (3.3-fold), *G. boreale* (11.3-fold), *H. nummularium* (24.5-fold), and *P. caeruleum* (3.8-fold) compared to the control. Similarly, for EP_28_, significant increases were observed for *G. boreale* (5.5-fold), *H. nummularium* (7.4-fold), and *P. caeruleum* (2.6-fold) (Table 2).

Although no statistically significant impact was observed in EP_14_ and EP_28_ due to NTP and stratification treatments for *C. glomerata*, both pretreatments significantly accelerated the emergence time in comparison to the control: compared to the control, the T_10_ for stratification was 1.9-fold shorter, and for NTP, it was 1.6-fold shorter. Similarly, the T_30_ for stratification was 2.5-fold shorter, while for NTP, it was 1.8-fold shorter in comparison to the control (Table 2). 

It is somewhat unexpected that stratification had a statistically significant adverse effect (*p* < 0.05) on the emergence of *F. vulgaris* seedlings, leading to a significant 2.4-fold decrease in EP_14_ and a 2.1-fold decrease in EP_28_ compared to the control (Table 2).

### 2.2. Assessment of Leaf Length

The influence of NTP and stratification pretreatments on growth parameters such as leaf length was investigated (Figure 2).

Seed pretreatments caused significant differences in leaf length for only 17.6% of the tested species. However, the findings of this study did not show any significant increase (*p* < 0.05) in leaf length due to the pretreatment effect in comparison to the control (Figure 2). Contrarily, a noticeable difference in leaf length was observed between the control group and both the NTP and stratification pretreatments in *S. ochroleuca*, indicating a statistically significant (*p* < 0.05) reduction under the pretreatments’ effects. The leaves of *S. ochroleuca* were 1.0-fold shorter under the stratification and 1.3-fold shorter under the NTP pretreatment compared to the control (Figure 2). The statistical test revealed significance (*p* < 0.05) in the means of leaf length for *C. glomerata* between NTP and stratification pretreatments: the leaves of *C. glomerata* were 1.3-fold longer under the NTP pretreatment. A contrary result was observed in the mean leaf length for *G. boreale* between NTP and stratification pretreatments: the leaves of *G. boreale* were 1.8-fold longer under the stratification pretreatment than NTP. We propose that the beneficial impact of stratification could be linked to the timing of sprout emergence, providing the sprouts with additional time to develop. When a certain number of sprouts emerge earlier, they have a longer period to grow. In general, pretreatments did not produce any significant differences in leaf length across these measurements in other species (Figure 2).

## 3. Discussion

Dormancy represents a physiological and biochemical mechanism serving multiple purposes, including the prevention of premature germination, the maintenance of seed viability over extended periods in soil and adverse environmental conditions, and the facilitation of seed dispersal [41,42]. Numerous studies have shown that NTP could be an efficient dormancy-breaking agent [29]. The impact of NTP treatment on seeds induces chemical modifications, alterations in charge, and structural changes, thereby enhancing agronomic seed quality through surface decontamination, improved germination, and the promotion of plant growth [29,43]. This study aimed to see how non-thermal plasma affects the emergence and growth of 17 plants from Lithuania’s meadows compared to cold stratification. However, the findings of this study only partially align with prior research that has shown the significant effects of NTP on seedling emergence, particularly as observed in studies involving cultivated plant species. For instance, prior studies have demonstrated that short-duration NTP treatments promote seed germination and enhance seedling growth in *A. thaliana* [28]. Similarly, NTP-pretreated tomato seeds have shown faster germination rates and significant improvements in growth parameters and seedling length [31]. Additionally, plasma pretreatment has been found to significantly enhance germination characteristics, vigor index, and seedling length in *D. carota* seeds [21]. Notably, NTP has also accelerated the germination rate of red clover seeds [30]. Furthermore, brief exposure to plasma has stimulated germination and reduced the mean germination time for wheat seeds [32].

Our findings indicate significant differences in seedling emergence trends for eight species: *A. vulneraria*, *D. grandiflora*, *F. vulgaris*, *G. boreale*, *H. nummularium*, *P. caeruleum*, *R. acris*, and *R. serotinus*. The most obvious finding to emerge from our analysis is the significant increase in seedling emergence for *A. vulneraria* under the NTP pretreatment. A significant positive increase (*p* < 0.05) in EP_28_ was observed, indicating a 4.5-fold increase compared to the control. While it is commonly recommended to subject *A. vulneraria* seeds to scarification as a pre-sowing treatment, the study conducted by Szalai et al. [44] presented an intriguing alternative. They found that subjecting the seeds to pre-chilling at −16 °C for a duration of 12 weeks, without scarification, stimulated a significant 35% increase in *A. vulneraria* seed germination. This observation highlights the necessity of stratification for this species. However, it also opens up the possibility of utilizing NTP as an additional method to enhance the final emergence of seedlings. The effects of NTP on plant species that exhibit physical seed dormancy, such as legumes, may hinge on alterations induced by NTP on the seed coat’s surface. These alterations can enhance the seed’s ability to absorb water and gases, as well as facilitate the release of germination inhibitors from the soaked seed [29]. In the current study, the favorable reaction of *A. vulneraria* to NTP aligns with similar outcomes observed with NTP pretreatment in other legume species. Previous studies have demonstrated that NTP treatment could reduce the hardness associated with the mechanical dormancy of many legume species (alfalfa, blue lupine, grass pea, honey clover, *Mimosa* sp., *Trifolium* sp., etc.) [45]. For instance, germination tests in vitro with the ‘Vyčiai’ red clover cultivar showed NTP pretreatments increasing germination rates by up to 24% [46].

Interestingly, when comparing EP_28_ values (percentage of seedlings emerged after 28 days) between different pretreatments, NTP exhibited a higher increase in EP_28_ for *A. vulneraria* but did not show statistically significant differences for other species, including *P. grandiflora.* This suggests that the positive effects of NTP on seedling emergence might be more prominent in the later stages of the experiment for certain species. In the context of *P. grandiflora*, the findings of our study indicate that NTP pretreatment primarily influenced the early phases of seedling emergence. While there was a significant positive effect on EP_14_ (a 1.4-fold increase compared to the control) and a reduction in T_10_ (1.5-fold compared to the control), these effects were not sustained through the later stages of emergence, as indicated by the absence of statistically significant differences in EP_28_. These findings are consistent with results obtained from seeds of another species within the *Lamiaceae* family, *Ocimum basilicum*. Previous research has shown that plasma pretreatments can indeed improve the germination potential of *O. basilicum* seeds [47]. Furthermore, NTP pretreatments have been shown to have positive effects on the phenotypic characteristics of *Salvia nemorosa* [48].

It is important to consider that the response to NTP pretreatment varies among different plant species, suggesting that the effectiveness of this method is influenced by the species’ specific physiological and ecological characteristics. For *P. grandiflora*, NTP pretreatment appears to expedite the initial phases of seedling emergence, which could be advantageous in specific growth and cultivation contexts. Numerous studies have reported that the effects of NTP treatment on germination were followed by similar effects on early seedling growth [30]. This study was able to demonstrate that regardless of the seed pretreatment method used, in general, seed pretreatment affected 3.5 times more species, inducing significant changes in the emergence percentage after 14 days compared to after 28 days.

However, it is crucial to note that the positive impact of NTP was not consistent across all plant species. Some species, such as *S. columbaria*, *D. grandiflora*, and *R. acris*, experienced a significant decrease in EP_14_ under NTP pretreatment: 2.1-fold, 10.0-fold, and 1.3-fold decreases, respectively, compared to the control. Moreover, in our study, NTP exhibited no significant effect on any of the emergence parameters for 58.8% of the tested plant species (Table 2). In general, our outcomes corroborate previous findings, suggesting that NTP may not exert a significant positive influence and, in some cases, could even negatively affect seedling emergence parameters for certain plant species. Neutral or negative effects were reported in other studies as well [30]. This underscores the importance of tailoring seed pretreatment methods to the specific needs and responses of target plant species, taking into consideration their unique physiological and ecological requirements. Furthermore, substantial intraspecies differences in the germination and emergence of seeds can exist even within a single plant species due to genetic polymorphism [49].

In the current study, comparing the two pretreatments, one of the important findings was that NTP caused increases in EP_14_ and EP_28_ for only 11.8% of the tested species, while stratification had a positive impact on 41.7% of them. Furthermore, when contrasting the effects of the two pretreatment methods, NTP and stratification, it became evident that stratification had a more pronounced influence on seedling emergence after 14 days. Stratification had a positive statistically significant impact on EP_14_ for *C. bononiensis* (3.3-fold), *G. boreale* (11.3-fold), *H. nummularium* (24.5-fold), and *P. caeruleum* (3.8-fold) compared to the control. However, EP_28_ showed lower significant increases in *G. boreale* (5.5-fold), *H. nummularium* (7.4-fold), and *P. caeruleum* (2.6-fold) under stratification compared to the control. Based on these data and the previously described findings, we can deduce that although NTP and stratification had a notable impact on specific species, the overall disparities in emergence became less prominent. This implies that the initial phases of germination exhibit higher sensitivity to pre-treatment methods, whereas the long-term emergence patterns tend to align.

When examining the impact of NTP and stratification on T_10_ and T_30_, some interesting patterns emerge. Stratification consistently resulted in significantly shorter times for seedling emergence compared to the control. This effect was particularly evident for species like *R. acris* and *S. columbaria*, where stratification outperformed both the control and NTP pretreatments (Table 2.) Additionally, stratification significantly reduced T_10_ and T_30_ for *C. glomerata* and T10 for *G. boreale*. 

In contrast, the emergence trend for *F. vulgaris* exhibited a distinct pattern. The dynamics of emergence of *F. vulgaris* seedlings under the stratification resulted in a significant 2.4-fold decrease in EP_14_ and 2.1-fold decrease in EP_28_ compared to the control.

Moving beyond seedling emergence, the study also investigated the influence of NTP and stratification on leaf length. Numerous studies provided evidence that the NTP treatment improved morphometric parameters of plants for various species [30]. Nonetheless, our study only partially confirms these findings. Although significant differences in leaf length were observed for only 17.6% of the tested species, the overall analysis did not indicate a significant increase in leaf length due to the pretreatment effect when compared to the control. However, it is worth noting that there were notable exceptions to this trend. A significant reduction in leaf length was observed for *S. ochroleuca* under both the NTP and stratification pretreatments (1.3-fold and 1.0-fold, respectively) compared to the control. Conversely, for *C. glomerata*, there was a significant increase in leaf length under the NTP pretreatment, with leaves being 1.3-fold longer compared to stratification. In contrast, *G. boreale* exhibited longer leaves under the stratification pretreatment, with leaves being 1.8-fold longer compared to NTP. It is hypothesized that the positive effect of stratification on leaf length may be linked to the timing of sprout emergence. Stratification appears to provide sprouts with more time to develop before emerging, potentially leading to longer leaves. Further research is needed to delve deeper into these dynamics and understand the underlying processes driving these effects.

Our findings in this study may be somewhat limited by the used methods. Various factors beyond the scope of this study could significantly influence the emergence of seedlings. The influence of seed pretreatment methods on germination and subsequent seedling emergence can vary significantly between different plant species and environmental conditions, as demonstrated by the contrasting results observed in sunflower and tomato seeds subjected to different pretreatments [50,51]. Further research is required to be undertaken to investigate the effects of these pretreatments under controlled laboratory conditions, such as Petri dishes, as well as in substrate environments.

In summary, our study provides valuable insights into the temporal progression of seedling emergence in response to different seed pretreatment methods for various wild plant species. We identified significant variations in emergence patterns, with some species benefiting from specific pretreatment methods. These findings emphasize the importance of tailoring seed pretreatment strategies to the individual requirements of plant species to optimize germination and seedling growth. Further research could explore the underlying mechanisms driving these differences. Furthermore, these findings underscore the significance of conducting extended observations under conditions that mimic the ecological characteristics of these species.

## 4. Materials and Methods

### 4.1. Seeds

The seeds of 17 wild plants were collected in natural habitats in Lithuania in 2020. The Table 3 below shows seed collection dates and areas in Lithuania. *C. bononiensis*, *D. borbasii*, *P. grandiflora*, and *V. pisiformis* were originally collected in nature in 2018 and then re-sown at the Lithuanian Research Centre for Agriculture and Forestry, Institute of Agriculture, and they were recollected for the experiment in 2020. A seed was defined as any disseminule, a part of a plant that serves to propagate, including botanical seeds with any additional tissues that assist dispersal, dry fruits, and fruit fragments according to ISMA [52]. The seeds were first air-dried and then grated by hand in order to break them. Sieving and flotation were used to clean the seeds. Flotation removed trash and empty, broken, or insect-damaged seeds. The clean seeds were then spread upon filter paper and left to dry. After drying, the seeds were stored at 18 °C in paper bags under dry conditions in the dark for two months. The experiment was conducted the following winter.

### 4.2. Experimental Design

The experimental design consisted of three distinct variants: a control group and two pretreatment conditions (NTP and stratification), all applied to 17 different species, each with four replications. Within each experimental variant, 200 seeds were employed and evenly distributed among four experimental units (pots), resulting in 50 seeds allocated for each replication (in the case of *S. columbaria*, 12 seeds were allocated for each replication). The experiment was conducted in a greenhouse at a temperature of +20 °C for a duration of 27 days, with a natural day and night light period, at the Lithuanian Research Centre for Agriculture and Forestry in December 2020. Control and pretreated seeds were placed in plastic pots with dimensions of 15 cm length × 15 cm width cm × 30 cm height filled with commercial peat characterized by the following properties: organic matter content > 80%, pH (EN 13037) between 5.5 and 6.5, electrical conductivity (EC) in mS/cm (1:1.5) between 0.5 and 0.8, and EC in mS/m (1:5, EN13038) of 20. During the experiment, the number of emerged sprouts was checked and counted 18 times. This experiment was based on a completely randomized design with three distinct variants, two of which involved seed pretreatments.

#### 4.2.1. NTP Pretreatment

In December 2020, seeds were pretreated with NTP at the Faculty of Natural Sciences, Vytautas Magnus University. The pretreatment was conducted using an atmospheric pressure scalable dielectric barrier discharge (DBD) device, manufactured at Kyushu University, Japan, with a homogeneous treatment area of 4 × 4.38 cm^2^ (Figure 3), as described in more detail elsewhere [53]. An electrical discharge was generated between wire electrodes separated by an insulating ceramic dielectric layer. Intact seeds were placed in a single layer on a glass sample holder underneath the electrode in a homogeneous zone. The discharge voltage, current, and power density were 7.0 kV, 0.2 A, and 3.1 W/cm^2^, respectively. The distance between the seed surface and the electrode was 5 mm. Seeds were exposed to NTP for 1 min. Seed pretreatment was conducted at room temperature, atmospheric pressure, and a relative humidity of air of 50–60%. Four days after the seed pretreatment with NTP, the emergence test started. This four-day duration for NTP seed pretreatment aligns with the methodology outlined in Mildazienė et al. [29].

#### 4.2.2. Stratification Pretreatment

Seeds were pretreated with stratification at the Lithuanian Research Centre for Agriculture and Forestry in December 2020. Seeds subsequently stored for 2 months after collection and drying were randomly arranged on three layers of filter paper in sterilized Petri dishes and watered with 2 mL distilled water. Then, the seeds were stratified in darkness at a constant temperature of 5 °C, with a relative humidity between 70% and 75%, for a period of 14 days. Seeds were provided with additional water in Petri dishes, if necessary, to prevent drying.

### 4.3. Leaf Length Measurements

Leaf length was evaluated within each experimental unit (pot) through the random selection of five individuals. In instances where there were five or fewer plant individuals present, all such individuals were included in the assessment. The sole objective of the measurements was to document the length of the longest leaves including leaflets. These measurements were conducted using a precision metal ruler calibrated to an accuracy of 1 mm. These observations were conducted during the month of January, corresponding to the growth stage of the examined species when it had reached the second principal growth stage, marked by the emergence of side shoots and the initiation of tillering.

### 4.4. Assessment of Seedling Emergence and Statistical Analysis

‘Survival analysis’ techniques were employed to manage interval-censored data associated with seedling emergence across all 17 plant species. In line with the methodology introduced by Onofri et al. [54], two nonparametric approaches were applied to estimate the CDFs. The primary method utilized was the NPMLE, which generates a stepwise curve representing the cumulative distribution of seed germination probabilities, relying on nonparametric maximum likelihood estimation. To assess the equality of time-to-event curves, a Wilcoxon exact test was conducted, providing permutation-based *p*-values, with a total of 199 permutations performed across randomization units (pots). When rank tests for NPMLE methods failed to detect significance in cases of intersecting time-to-event curves [40], an alternative nonparametric approach was employed, involving the calculation of the CDF using the KDE statistic. To enhance bandwidth selection, the asymptotic mean integrated squared error (AMISE) method proposed by Barreiro-Ures et al. [55] was utilized. Subsequently, differences in seedling emergence trends were assessed by calculating the Cramér–von Mises-type distance statistic [54,55].

To assess the relative rankings of the experimental pretreatments concerning the speed of species seedling emergence, we calculated the T_10_ and T_30_ indexes. These indexes were derived by determining the 10th and 30th percentiles of the time-to-event distribution using NPMLE method. To establish their 95% confidence intervals (CIs), we utilized a resampling technique, performed at the level of the experimental unit ([54] different letters indicate non-overlapping 95% confidence intervals for T_10_ and T_30_).

All the time-to-event data-related methods mentioned above were executed using the ‘drcte’ package in R [56].

Emergence percentages after the 14th day of the experiment were calculated by the formula:(1)$${EP}_{14}=\frac{n_{14}}{N}\times 100\%$$

In this formula, ‘$n_{14}$’ represents the total number of seedlings that germinated within a single experimental unit by the 14th day of the experiment; and ‘*N*’ denotes the total number of seeds used per one experimental unit (pot).

Subsequently, the calculation provided an estimation of the emergence percentage on the final day of the 28-day experiment using the formula:(2)$${EP}_{28}=\frac{n_{28}}{N}\times 100\%$$

Here, ‘$n_{28}$’ signifies the overall number of seedlings that germinated within a single experimental unit on the final (28th) day of the experiment. ‘*N*’ denotes the total number of seeds used per one experimental unit (pot).

Measurements of seedling leaf lengths did not require any transformations; however, before testing for statistical differences in proportion data (emergence percentages), a transformation was applied using the arcsine square root method. If all assumptions of data normality and homoscedasticity were met, significant differences in means were assessed through one-way analysis of variance (ANOVA), followed by Tukey’s multiple comparison analysis. However, if the Shapiro–Wilk test confirmed a non-normal distribution in the data, the nonparametric Kruskal–Wallis test for the equality of medians was employed. In cases where significant differences were detected, Dunn’s post hoc test was utilized. Different letters (a–c) denote statistically significant differences at a significance level of *p* < 0.05. The 95% confidence intervals (CIs) provided were obtained using the adjusted percentile method (BCa) with 99,999 bootstrapping replicates. This method ensures a robust estimation of the confidence intervals despite potential deviations from normality.

The statistical methods mentioned earlier were applied using the computer program PAST version 4.13 [57].

## 5. Conclusions

In summary, our study examined the effects of NTP and cold stratification on 17 perennial plant species from Lithuanian meadows. Statistically significant differences in seedling emergence were observed, with stratification having a more pronounced positive impact on 41.7% of the species. Observations from this study have shown that NTP treatment led to increases in both EP_14_ and EP_28_ for a mere 11.8% of the examined plant species, whereas stratification demonstrated a beneficial effect on a substantial 41.7% of these species. Evidently, NTP exhibited a positive impact on the emergence of *P. grandiflora* after 14 days and *A. vulneraria* after 28 days, while it had adverse effects on *S. columbaria*, *D. grandiflora*, and *R. acris*. Stratification, on the other hand, significantly increased emergence for *C. bononiensis*, *G. boreale*, *H. nummularium*, and *P. caeruleum* but decreased it for *F. vulgaris*. NTP showed a notable positive effect on the emergence of *A. vulneraria* seedlings, with a 4.5-fold increase in emergence after 28 days. This suggests the potential utility of NTP as an additional method to enhance the seedling emergence of certain species. However, the impact of NTP was not consistent across all species, and in some cases, it even resulted in significantly reduced seedling emergence.

Furthermore, our study explored the effects of NTP and stratification on leaf length. Although significant differences were observed in only 17.6% of the species, the overall analysis did not indicate a significant increase in leaf length due to pretreatment. *S. ochroleuca* had reduced leaf length under both NTP and stratification, while *C. glomerata* had longer leaves under NTP and *G. boreale* had longer leaves under stratification.

This study highlights the need to customize pretreatment methods to specific plant species and to conduct further research to understand the underlying mechanisms. Controlled laboratory studies and substrate environments should also be explored. These findings have implications for optimizing germination strategies and enhancing seedling establishment in grassland restoration and conservation efforts.

## Figures and Tables

**Figure 1 plants-12-03220-f001:**
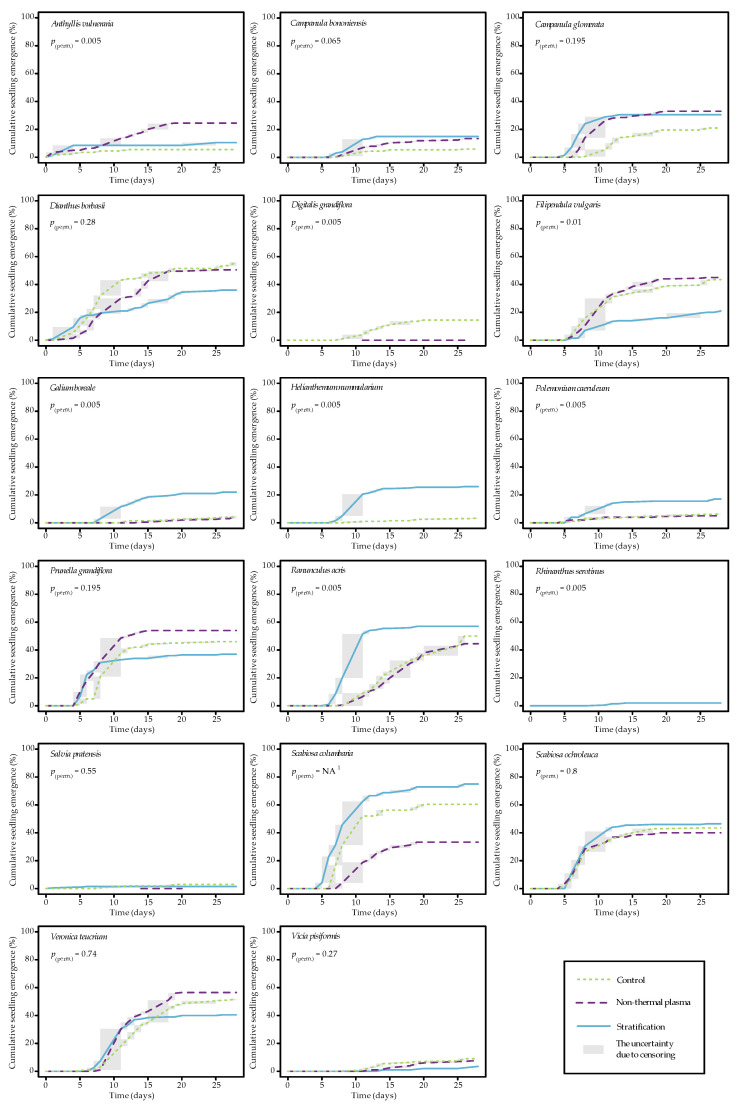
Nonparametric time-to-event curves based on nonparametric maximum likelihood estimation (NPMLE) for seedling emergence assay of 17 plant species. P*_perm_* values from the Wilcoxon exact test are displayed. Note: ^1^ NA—in the case of *Scabiosa columbaria*, the P*_perm_*_._ value is not available due to the use of *N* = 12 seeds per experimental unit (pot), *N* = 50 seeds were used for all other species.

**Figure 2 plants-12-03220-f002:**
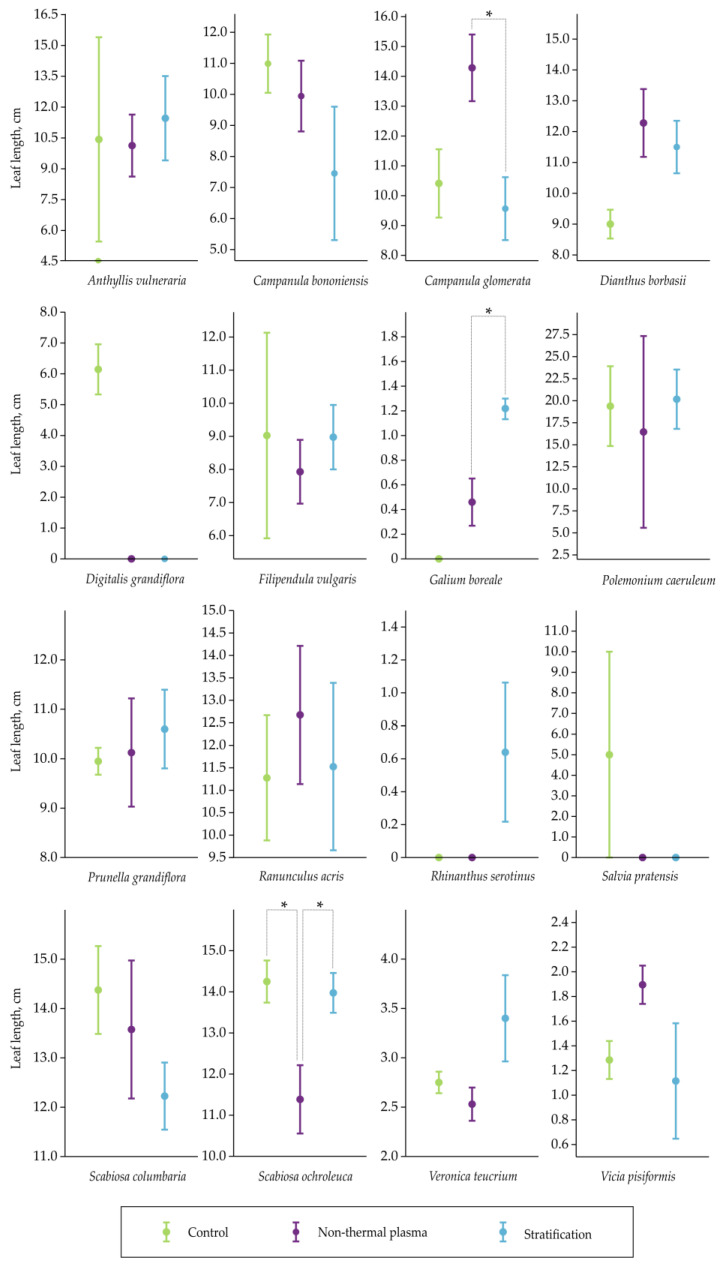
Seedling leaf length comparison—mean values with associated standard errors. Averages were calculated from each of four experimental units (pots) per experimental variant. One-way ANOVA and Tukey’s post hoc test assessed mean equality among experimental variants for each species separately. Statistically significant differences (*p* < 0.05) are indicated with an asterisk (*).

**Figure 3 plants-12-03220-f003:**
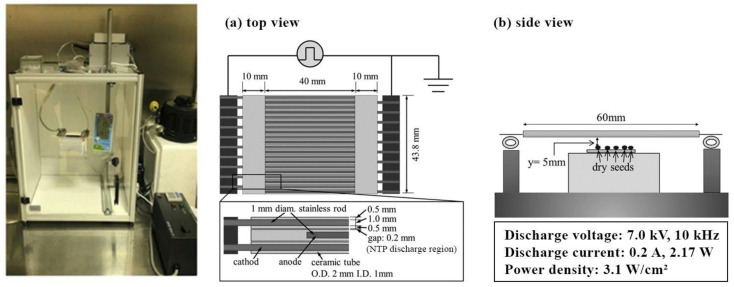
Experimental device and setup of the dielectric barrier discharge plasma device for seed pretreatment with NTP.

**Table 1 plants-12-03220-t001:** Germination responses of selected plant species to different pretreatment methods. Presented data are sourced from Seed Information Database ENSCOBASE [3].

Plant Species	No Pretreatment	Scarification	Cold Stratification
*Anthyllis vulneraria*	100% (MTG 8 d.)	100% (duration 5 d.)	
*Campanula bononiensis*	86.7% (duration 63 d.) GA3		
*Campanula glomerata*	8% (duration 27 d.), 95% (MTG 10.4 d.) GA3	75% (MTG 26.9 d.) GA3	
*Dianthus borbasii* (data for *D. carthusianorum*)	100% (duration 42 d., MTG 7.4 d.)		91% (duration unknown)
*Digitalis grandiflora*	94% (MTG 9.7 d.)		99% (duration unknown) GA3
*Filipendula vulgaris*	100% (duration unknown), 95% (MTG 14.7 d.)		100% (MTG 7 d.)
*Galium boreale*	90% (MTG 16.7 d.) GA3	65% (MTG 12.9 d.)	
*Helianthemum nummularium*		100% (duration 6 d.)	
*Polemonium caeruleum*	89% (MTG 11.1 d.)		100% (MTG 7 d.) GA3
*Prunella grandiflora*	*P. vulgaris* 100% (MTG 9.8 d.)		
*Ranunculus acris*	98% (MTG 17.6 d.)		
*Rhinanthus serotinus* (data for *R. minor*)	100% (MTG 52.6 d.)		100% (MTG 1.4 d.)
*Salvia pratensis*	100% (MTG 7 d.)		
*Scabiosa columbaria*	79% (MTG 17.3 d.)		79% (MTG 14.5 d.)
*Scabiosa ochroleuca* (data for *S. columbaria*)	79% (MTG 17.3 d.)		79% (MTG 14.5 d.)
*Veronica teucrium* (data for *V. prostrata*)	90% (duration 62 d.)		
*Vicia pisiformis*	(data for *V. dumetorum*) 35% (duration unknown)	(data for *V. sylvatica*) 100% (duration unknown)	

MTG—the mean time to germinate, GA3—chemical treatment using gibberellic acid.

**Table 2 plants-12-03220-t002:** Evaluation of seedling emergence parameters for 17 plant species.

Species	Pretreatment	EP_14_ *, %	EP_28_ *, %	T_10_, Days	T_30_, Days
*Anthyllis vulneraria*	C	5.5 (1.0–9.0) a	5.5 (1.0–9.0) a	9.2 (7.3–12.0) a	–
	N	17.5 (12.0–23.0) a	24.5 (17.0–31.0) b	9.6 (7.0–13.5) a	17.0 (14.0–19.0) nt
	S	8.5 (3.5–12.0) a	10.5 (8.5–12.0) a	12.4 (2.8–25.0) a	–
*Campanula bononiensis*	C	4.5 (2.5–5.5) a	6.0 (4.5–7.0) a	16.8 (12.9–26.0) a	–
	N	9.5 (8.0–11.0) a	13.5 (9.5–16.0) a	16.0 (10.7–25.6) a	–
	S	15.0 (8.0–20.0) b	15.0 (8.0–20.0) a	10.1 (8.9–12.3) a	12.5 (12.5–12.5) nt
*Campanula glomerata*	C	14.5 (10.0–17.5) a	21.0 (12.0–28.0) a	12.4 (10.7–16.2) a	24.2 (17.5–26.0) a
	N	28.5 (19.5–34.5) a	33.0 (22.0–41.0) a	7.7 (7.0–9.1) b	13.5 (9.7–19.3) b
	S	30.5 (20.0–39.0) a	30.5 (20.0–39.0) a	6.4 (5.8–7.1) b	9.6 (7.5–12.9) b
*Dianthus borbasii*	C	45.0 (35.0–54.0) a	56.0 (46.0–64.0) a	5.0 (3.7–6.5) ab	8.4 (6.6–11.4) a
	N	37.0 (24.0–44.0) a	50.5 (32.0–36.5) a	6.4 (5.3–7.6) a	11.8 (8.9–15.8) ab
	S	23.5 (19.0–27.0) a	36.0 (34.0–38.0) a	4.1 (3.0–5.4) b	17.9 (13.0–25.1) b
*Digitalis grandiflora*	C	10.0 (3.5–15.5) a	14.5 (9.0–20.0) a	14.8 (11.5–20.0) a	–
	N	1.0 (0.0–1.5) b	12.0 (8.5–15.5) a	25.7 (25.5–26.0) b	–
	S	0 nt	0 nt	–	–
*Filipendula vulgaris*	C	33.0 (28.5–37.0) a	43.5 (40.0–47.0) a	7.1 (6.1–8.0) a	13.1 (10.4–19.1) a
	N	36.0 (30.0–40.0) a	45.0 (40.0–49.0) a	7.9 (7.0–8.7) a	11.6 (10.0–14.7) a
	S	14.0 (10.0–16.0) b	21.0 (16.5–24.5) b	10.8 (7.7–19.0) a	27.1 (25.6–27.7) b
*Galium boreale*	C	1.5 (0.5–2.0) a	4.0 (2.0–5.5) a	25.8 (25.6–26.0) a	–
	N	0.5 (0.0–1.0) a	5.0 (2.0–7.0) a	27.8 (27.5–28.0) b	–
	S	17.0 (11.0–22.0) b	22.0 (14.5–27.0) b	10.9 (8.7–14.4) c	20.9 (14.6–26.0) nt
*Helianthemum nummularium*	C	1.0 (0–1.5) a	3.5 (2.0–4.0) a	–	–
	N	0.5 (0.0–1.0) a	0.5 (0.0–1.0) a	–	–
	S	24.5 (16.5–35.5) b	26.0 (18.0–33.0) b	9.2 (8.1–10.8) nt	14.2 (10.3–25.9) nt
*Polemonium caeruleum*	C	4.0 (2.0–5.0) a	6.5 (4.5–7.5) a	23.6 (16.6–28.0) ab	–
	N	4.0 (2.5–5.0) a	5.0 (2.0–7.0) a	23.0 (18.5–25.0) a	–
	S	15.0 (9.0–18.0) b	17.0 (12.0–20.0) b	10.8 (7.2–26.3) b	–
*Prunella grandiflora*	C	42.0 (36.0–48.0) a	46.0 (40.5–50.5) ab	7.4 (7.1–7.6) a	9.7 (8.4–11.1) a
	N	53.0 (49.0–57.0) b	54.0 (48.5–58.5) b	5.1 (4.7–5.8) b	7.8 (6.8–9.0) a
	S	34.0 (32.5–35.5) a	37.0 (36.0–37.5) a	5.3 (4.8–5.8) b	9.4 (5.9–19.2) a
*Ranunculus acris*	C	21.5 (18.0–25.5) a	50.0 (48.0–51.0) ab	11.5 (10.1–13.0) a	17.3 (14.4–21.3) a
	N	16.0 (8.5–22.0) b	44.5 (39.5–48.0) b	12.3 (10.2–14.4) a	18.1 (15.4–20.8) a
	S	55.5 (48.0–64.0) ab	57.0 (51.0–63.0) a	7.2 (6.7–7.7) b	9.0 (8.2–9.8) b
*Rhinanthus serotinus*	C	0 nt	0 nt	–	–
	N	0 nt	0 nt	–	–
	S	2.0 (0.0–3.0) nt	2.0 (0.0–3.0) nt	–	–
*Salvia pratensis*	C	2.0 (0.5–3.0) a	3.0 (2.0–3.5) a	–	–
	N	1.0 (0.0–1.5) a	1.5 (0.0–2.5) a	–	–
	S	1.5 (0.0–3.0) a	1.5 (0.0–3.0) a	–	–
*Scabiosa columbaria*	C	56.2 (39.6–66.7) a	60.4 (39.55–75.0) ab	6.7 (6.4–7.7) a	8.2 (7.0–10.5) ab
	N	27.1 (18.8–31.2) b	33.3 (27.08–37.5) b	9.5 (7.9–12.4) a	14.6 (10.5–18.7) a
	S	68.8 (54.2–79.2) a	75.0 (62.48–83.4) a	5.4 (4.9–5.9) b	6.8 (5.7–8.3) b
*Scabiosa ochroleuca*	C	39.0 (26.0–50.0) a	43.5 (35.5–50.0) a	6.4 (5.9–6.9) a	10.3 (7.5–16.6) a
	N	37.0 (31.0–42.0) a	40.0 (35.0–43.0) a	6.2 (5.4–7.1) a	9.6 (7.4–14.8) a
	S	45.5 (36.0–53.0) a	46.5 (36.5–51.5) a	6.0 (5.6–6.5) a	8.2 (7.2–10.5) a
*Veronica teucrium*	C	33.0 (23.0–43.0) a	52.0 (41.5–61.0) a	9.4 (8.7–10.4) a	13.7 (11.7–16.8) a
	N	41.0 (25.0–52.0) a	56.5 (44.0–64.5) a	9.0 (8.7–9.4) a	11.5 (10.1–15.6) a
	S	37.5 (29.5–44.0) a	40.5 (32.5–44.5) a	8.3 (7.7–8.9) a	11.5 (10.1–14.7) a
*Vicia pisiformis*	C	5.5 (2.5–8.0) a	9.0 (6.0–12.0) a	22.9 (13.9–26.0) a	–
	N	1.5 (0.0–2.5) a	7.5 (4.0–11.0) a	23.4 (17.2–27.0) a	–
	S	1.0 (0.0–1.5) a	3.5 (0.5–6.0) a	–	–

* EP_14_, EP_28_—emergence percentage after 14 and 28 days, respectively. The “nt” designation signifies untested conditions. In the columns EP_14_ and EP_28_, lowercase letters indicate outcomes derived from one-way ANOVA and subsequent Tukey’s pairwise post hoc assessments, employing arcsine-transformed data. In instances where the presumptions of data normality and comparable variances are notably contravened, recourse was made to nonparametric Kruskal–Wallis analysis accompanied by Dunn’s post hoc procedure. For *Digitalis grandiflora* exclusively, the *t* test for equal means was implemented. Different letters indicate non-overlapping 95% confidence intervals for T_10_ and T_30_.

**Table 3 plants-12-03220-t003:** List of the studied plant species, with indication of their seed collection dates and areas in Lithuania.

Plant Species	Location Geographical Coordinates **	Seed Collection Dates
*Anthyllis vulneraria*	Sibirka, Trakai distr. 54.667517, 24.899277	11 August 2020
*Campanula bononiensis* * ^EN^	Dotnuva, Kėdainiai distr. 55.396349, 23.865125	21 August 2020
*Campanula glomerata*	Krekenava, Panevėžys distr. 55.540838, 24.111416	21 August 2020
*Dianthus borbasii* * ^EN^	Dotnuva, Kėdainiai distr.	21 August 08 2020
*Digitalis grandiflora*	Sibirka, Trakai distr. 54.667517, 24.899277	11 August 08 2020
*Filipendula vulgaris*	Bradeliškės, Vilnius distr. 54.82638, 24.949936	1 October 2020
*Galium boreale*	Bradeliškės, Vilnius distr. 54.82638, 24.949936	1 October 2020
*Helianthemum nummularium*	Bradeliškės, Vilnius distr. 54.82638, 24.949936	4 October 2020
*Polemonium caeruleum* * ^VU^	Bradeliškės, Vilnius distr. 54.82638, 24.949936	1 October 2020
*Prunella grandiflora* * ^EN^	Dotnuva, Kėdainiai distr. 55.396349, 23.865125	21 August 2020
*Ranunculus acris*	Liubavas, Vilnius distr. 54.85014, 25.32080	10 October 2020
*Rhinanthus serotinus*	Rusnė, Šilutė distr. 55.3252, 21.455061	30 September 2020
*Salvia pratensis* * ^VU^	Jurbarkas distr. 55.08005, 23.40588	3 November 2020
*Scabiosa columbaria*	Vilnius distr. 54.91008, 25.32236	23 August 08 2020
*Scabiosa ochroleuca*	Sibirka, Trakai distr. 54.667517, 24.899277	2 October 2020
*Veronica teucrium*	Krekenava, Panevėžys distr. 55.540838, 24.111416	21 August 2020
*Vicia pisiformis* * ^NT^	Dotnuva, Kėdainiai distr. 55.396349, 23.865125	15 July 2020

* Species’ threat category according to IUCN Red List: EN—Endangered, VU—Vulnerable, NT—Near Threatened [37]. ** The geographic coordinates of locations are presented in Decimal Degree format for North latitude and East longitude, using the World Geodetic System 1984 (EPSG code 4326) as the reference datum.

## Data Availability

The data presented in this study are openly available in The National Open Access Research Data Archive (MIDAS) at https://doi.org/10.18279/MIDAS.TheImpactofNTPonWildPlantSeeds.209356, accessed on 28 August 2023, reference number [58].

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
