# Peer review of "Comparing Non-Thermal Plasma and Cold Stratification: Which Pre-Sowing Treatment Benefits Wild Plant Emergence?"

_plants, 2023, doi:10.3390/plants12183220_

Round 1
Reviewer 1 Report
Non-thermal plasma (NTP) technologies are being developed for agriculture and shown the potential to improve seed quality by enhancing germination and promoting plant growth. This manuscript evaluate 17 plant seeds and compare it with the moist stratification pretreatment. The results obtained indicate that there were differences in emergence parameters among the species and NTP did not have significant differences for the control. This result is very interesting and has some research assistance.
Generally, evaluating leaf length alone is not sufficient for plant phenotype analysis.such as changes in plant photosynthesis rate, total biomass, etc. are necessary.
Moderate editing of English language required
Author Response
We would like to thank you for editing the article and the time dedicated to it. We have made significant revisions based on your comments.
Please see the attachment.

Reviewer 2 Report
The manuscript presents results of non-thermal plasma (NTP) and stratification treatment of seeds of 17 wild plant species originating from meadows in Lithuania. In a search for new practice of restoring biodiversity of meadow plant species, the study aimed to evaluate the effectiveness of NTP treatment compared to the standard stratification method. Authors present different parameters of seedling emergence and seedling growth after specific type of pre-sowing treatment. In general, the manuscript is properly organized. However, some important corrections of sections of the manuscript must be made before publishing.
Abstract:
Line 22: The previously introduced shortcut for non-thermal plasma (NTP) should be used here instead of the full form.
Introduction:
Line 43: The abbreviation for “non-thermal plasma” should be introduced at its first appearance in the main text.
Line 50: “Vigor index” should be written as “vigor index”. There is no need to use capital letter. This correction should be followed throughout the manuscript.
Line 51-52: The reference no. 25 does not refer to NTP treatment. It should be replaced.
Line 55-56: The sentence, “The rate of germination of red clover seeds in the control group was considerably faster” does not indicate a stimulatory effect of NTP treatment.
Line 64: The sentence should be revised as follows: “…by the short period of seedling emergence observation...”
Line 65: For better clarity, I suggest the use of “seedling emergence” instead of “seed emergence” throughout the manuscript.
Line 69: The cited reference no. 31 is not Kornarzynski et al.
Line 70: It should be: “Liu et al” instead of “Liu Bo's”.
Line 70-71: The sentence in lines 70-71 is not clear. It should be revised.
Line 76-77: The plant classes introduced here should be introduced also in Table 1 for better clarity.
Line 82-83: Again, for better clarity I suggest to indicate the protected species in Lithuania in Table 1 (for example with the red colour).
Line 85: Please recheck and correct the citation of the references no. 40 and 41.
Results:
In results description, there are several inaccuracies and errors in relation to the data presented in the tables and figures. When describing the results, the Authors should refer to the appropriate numbers of tables and figures. Moreover, changes without statistical significance are further commented in many places. Statistically not significant effect implies the absence of such effect. Therefore, comments regarding effect changes must only refer to statistically significant results. Example of such incorrect comment is in line 170-171: “The stratification also had a positive effect on Helianthemum nummularium, but it did not differ significantly from the control group”. The whole Results section should be carefully revised to use only statistically significant effect changes from the control or other treatment when referring to positive/ negative treatment outcomes.
Other specific comments:
Line 101: The Two-way ANOVA is not included in Materials and Methods section.
Line 118-121: The Authors should specify here whether the described fragment of the results refers to the number of specific plant species or specific parameters of emergence.
Line 122: What does “early sowing” imply?
Line 123: Should be: “The NTP treatment showed a significantly positive effect (53%) …”
Line 136: For clarity the sentence should be revised as follows: “The highest values of final emergence percentage (more than 50%) was observed for…”
Line 138: Should be: “Dianthus borbasii (56% under control)”.
Line 139-142: I suggest to arrange the order of the listed plant species from the smallest to the largest values of the analysed parameter (respectively to data in Table 3).
Line 142-144: The results described here are not statistically significant (Table 3).
Line 145: Should be: “Significant differences in final seedling emergence were found between stratification (75%), and NTP (33.3%) treatments in Scabiosa columbaria, although stratification and the control (60.4%) did not differ.
Line 148-149: Should be: “…while significantly negative effects were observed under stratification compared to NTP treatment.”
Line 156: Should be “Filipendula vulgaris” instead of “Digitalis grandiflora” (Table 3).
Line 160-162: The context of the statement, “the joint seedlings’ emergence percent mean of 5 species” is not clear.
Line 162: I suggest to use “protected” instead of “threatened”.
Line 168: The sentence should be revised. What refers to “positive effects”? Moreover, the explanation for the presented values (27.9% and 10.6%) should be given.
Line 170-171: Results that are not statistically significant should not be commented as a positive/negative effect. This correction should be implemented throughout the manuscript.
Line 172: Recheck the given species name according to data presented in Table 3.
Line 174: The explanation for the presented numbers (49.1% and 34.5%) should be provided in the text.
Line 174: Again, recheck and correct the description according to data presented in Table 3.
Line 178: MGT abbreviation should be explained at its first appearance.
Line 181: The sentence should be revised as follows: “…reached the lowest value of EI parameter with a result of 4.1 days for emergence…”
Line 181-183: The sentence should be revised as follows: “However, several species achieved the value of EI index lower than 10 days: Campanula bononiensis – 9.8 days under stratification; Campanula glomerata – 7.9 days under stratification …”
Lines 187-217: This fragment of results should be rewritten. According to data presented in Table 3, there is no statistical analysis for EI, T10 and T30 parameters, therefore the positive or negative effects can’t be identified. Moreover, the numerical values included in the text should be each time adequately and clearly described (not in short form). Also, the presented data cannot be described selectively.
Line 217-222: Which data (table, figure) does this description applies to?
Line 224-225: The sentence should be revised as follows: “… Ranunculus curves and Anthyllis vulneraria show that there was significant difference between methods of seed treatment.
Line 225-230: The sentences are not clear. More clarification should be given. Which data (table, figure) does this description applies to?
Line 235-260 and 269-278: These sections should be revised to clearly indicate which results are statistically significant and only such effects should be referred to as positive or negative compared to specific controls. In the text, references to specific figures with relevant data should be given.
Line 265: Should be “NTP” instead of “NT plasma”. Moreover, clarify if stems or leaves were measured according to details given in subsection 4.3 of Materials and Methods.
Line 279-285: A reference to proper Table or Figure should be given for this description. Recheck data given with particular numbers (line 283).
Discussion:
In this section, corrections for grammar and style are necessary in many sentences.
Line 290: “NTP” abbreviation should be used. The same in line 297, 309, 399.
Line 302: Reference no. 34 presents results of a tomato study, not for Arabidopsis thaliana.
Line 304. Reference no. 56 does not appear in the bibliography.
Line 305-307: Did the Authors mean the “treated group” instead of the “control group”?
Line 315-318: The sentence should be revised as follows: “Although Anthylis vulneraria pre-sowing scarification treatments is recommended, in the case of the study by Szalai et al., pre-chilling at -16°C for 12 weeks without scarification stimulates 35% of Anthyllis vulneraria seed germination [44] and proves stratification to be necessary, but …”
Line 331: Should be: “the percent of final seedlings emergence”.
Line 332: Should be: “observation time/ period”
Line 335-338: The sentence is not clear and should be revised.
Line 340-342: The sentence is not clear and should be revised. The given data (56%) should be rechecked.
Line 349-350: Comments must focus on statistically significant results only.
Line 354: The statement “NTP negatively affects Scabiosa columbaria” is not clear. Please clarify.
Line 354: The abbreviation “GRI” should be explained at first appearance.
Line 358: Change the statement “about Liu Bo experiment” to “with study of Liu et al.”
Line 361-367: Comments must be focused on statistically significant results only.
Line 370-378: The sentences should be revised for better clarity.
Line 387-389: The sentence is not clear and should be revised. Are the statements here supported by presented data?
Line 394-395: Should be “Equally, the results of Chen et al. suggest that …”
Materials and Methods:
Line 427: The term “cold plasma” appears here for the first time. To avoid any confusion, I suggest to use only “NTP” term throughout the manuscript when referring to own study.
Line 431: The reference no. 3 is not related to the details of NTP treatment.
Line 435-436: For how long was the NTP treatment performed?
Line 437: The sentence: “Seeds were exposed to the non-thermal plasma treatment as described above.” is not contextually clear here.
Line 438: “CP treatment” should be changed to “NTP treatment”. Why were seeds sown in pots 4 days after treatment? What were the dimensions of the pots?
Line 439: What kind of turf was used (composition, manufacturer)?
Line 441-442: The stratification method should be described in more details. Throughout the manuscript the term “moist stratification” is often used. Therefore, it should be well described in Materials and Methods section and the term for stratification should be uniformed throughout the manuscript. How much water was used in Petri dishes during stratification? Was the stratification performed in dark conditions?
Line 446: “CP” should be replaced with “NTP”.
Line 449: The description of phenotypic traits analysis is confusing. In section 4.3 Authors state that the stem length was measured, but in Results section leaf length is given (line 266; Figure 2; also line 483 in Materials and Methods section). Moreover, if the measured plants were in the stage of tillering, was the main stem/leaf analysed?
Line 458: Should be: “seedling emergence” instead of “seedling emergency”. Please correct this error throughout the manuscript
Line 461: “Germinated” should be replaced with “emerged”. This should be applied also in line 467.
Line 468: “N” from formula (2) should be explained.
Line 469: The abbreviation “ETI” used in Result section should be introduced here for the Emergence Rate Index. The same should be done for Emergence Index (EI) in line 475.
Line 473: “D” should be lowercase “d”. Explanation for “N” from formula (3) is missing.
Line 479, 485: The explanations for symbols in formula (4) and (5) should be given.
Moreover, the methods for calculating T10 and T30 factors, which are presented in the Results section (i.a. Table 3) are not included in Materials and Methods section.
Line 502: “Survival analysis” should be in quotation marks.
Line 519: According to reference no. 55 “drcte” should be written in uppercase “DRCTE”.
Conclusions:
Line 522: Should be “NTP” instead of “NT plasma”.
References:
Line 625: Reference no.26 contains two articles.
In Discussion section, corrections for grammar and style are necessary in many sentences.
Author Response

(The authors gave the same response as above.)

Round 2
Reviewer 1 Report
It is OK now.